# Identification of Messenger RNA Signatures in Age-Dependent Renal Impairment

**DOI:** 10.3390/diagnostics13243653

**Published:** 2023-12-13

**Authors:** Katsunori Yanai, Shohei Kaneko, Akinori Aomatsu, Keiji Hirai, Susumu Ookawara, Yoshiyuki Morishita

**Affiliations:** 1Division of Nephrology, First Department of Integrated Medicine, Saitama Medical Center, Jichi Medical University, Saitama 330-8503, Japan; shohei.sasurai@gmail.com (S.K.); ayksera@gmail.com (A.A.); keijihirai@kfy.biglobe.ne.jp (K.H.); su-ooka@hb.tp1.jp (S.O.); ymori@jichi.ac.jp (Y.M.); 2Division of Intensive Care Unit, First Department of Integrated Medicine, Saitama Medical Center, Jichi Medical University, Saitama 330-8503, Japan

**Keywords:** messenger RNA, age-dependent renal impairment, qRT-PCR, microarray

## Abstract

In general populations, age-dependent renal impairment contributes to the progression of renal dysfunction. It has not been known which molecules are involved in age-dependent renal impairment. Messenger RNA (mRNA) has been reported to modulate various renal diseases, and we therefore investigated mRNA signatures in age-dependent renal impairment. We performed an initial microarray-profiling analysis to screen mRNAs, the expression levels of which changed in the kidneys of 50-week-old senescence-accelerated prone (SAMP1) mice (which have accelerated age-dependent renal impairments) compared with those of 50 wk old senescence-accelerated-resistant (SAMR1) mice (which have normal aged kidneys) and with younger (10 wk old) SAMP1 and SAMR1 mice. We next assessed the expressions of mRNAs that were differentially expressed in the kidneys of SAMP1-50wk mice by conducting a quantitative real-time polymerase chain reaction (qRT-PCR) and compared the expressions among the SAMP1-10wk, SAMR1-10wk, and SAMR1-50wk mice. The results of the microarray together with the qRT-PCR analysis revealed five mRNAs whose expression levels were significantly altered in SAMP1-50wk mouse kidneys versus the control mice. The expression levels of the five mRNAs were increased in the kidneys of the mice with age-dependent renal impairment. Our findings indicate that the five mRNAs might be related and could become therapeutic targets for age-dependent renal impairment.

## 1. Introduction

Age-dependent renal impairment frequently occurs among individuals with end-stage renal disease (ESRD) [1]. Many genetic and environmental factors have been observed to be associated with age-dependent renal impairment and its progression, e.g., salt intake, lead exposure, tobacco use, genetic susceptibility, thrombophilia, low birth weight, and metabolic syndrome [2]. However, the pathogenesis of age-dependent renal impairment, including the precise underlying molecular mechanisms, has been unclear. Aging-related arterial intimal fibrosis has been reported to cause ischemia in aging kidneys, and this results in glomerulosclerosis that contributes to the initiation of age-dependent renal impairment [3,4]. Tubulointerstitial changes can also occur due to age-associated arteriosclerosis and the subsequent hypoxia/ischemia in the kidneys’ medulla and cortex, leading to the development of age-dependent renal impairment [5].

Messenger RNAs (mRNAs) are known to moderate the transfer of genetic information from cell nuclei to the ribosomes in the cytoplasm, where the information provides a template for protein synthesis [6]. Pivotal roles of mRNAs in the progression of inflammation, cancer, and a large number of other pathological or physiological conditions have been described [7,8,9], and mRNA has shown a therapeutic potential in cancer, infectious diseases, and viral vaccines [7,8,9,10]. The relationship between mRNAs and age-dependent renal impairment is not yet understood, and the possibility of applying mRNAs in treatments for age-dependent renal impairment has not been investigated to our knowledge. We conducted the present study and screening using a murine model to identify mRNAs that may be involved in age-dependent renal impairment. Normal strains of mice generally do not experience significant age-dependent renal impairment, and, in this study, we applied a murine model, i.e., senescence-accelerated mouse-prone (SAMP1) mice, which experience age-dependent renal impairment. The control group was a strain that undergoes normal aging: senescence-accelerated mouse-resistant (SAMR1) mice. To investigate the development of the mechanism of age-dependent renal impairment, we conducted this study using these mice models.

## 2. Methods and Materials

### 2.1. Ethical Approval

The Ethics Committee of Jichi Medical University (Saitama, Japan) approved this study. The university’s Animal Ethics Committee approved the experimental protocol, which we conducted according to the guide on the use and care of laboratory animals issued by the University’s Experimental Animal Committee.

### 2.2. The Age-Dependent Renal Impairment Mouse Model

Male 10-week-old SAMP1 mice and male 10-week-old SAMR1 mice (weights 25–30 g) were obtained from CLEA Japan (Tokyo, Japan), housed under antiviral antibody-free micro-isolation conditions, and given standard chow ad libitum for the 50-week study period. Each group was allocated four mice.

### 2.3. Microarray Analysis

Hokkaido System Science (Hokkaido, Japan) performed the analyses of the expressions of mRNAs by using a Low Input Quick Amp Labeling Kit (Agilent Technologies; Santa Clara, CA, USA) in accordance with the manufacturer’s instructions. First, 50 ng of total RNAs was denatured using T7 Promotor Primer at 65 °C for 10 min. The specimens were then chilled on ice for 5 min. We then incubated the specimens using cDNA master mix at 40 °C for 2 h. The enzymes of the specimens were next inactivated at 70 °C for 15 min. We then chilled the specimens on ice for 5 min and incubated using transcription master mix at 40 °C for 2 h. We labeled each specimen by using the Qiagen (Valencia, CA, USA) RNeasy Mini Kit per the manufacturer’s instructions.

Next, we hybridized the specimens on Agilent Technologies 8 × 60 K SurePrint G3 mouse miRNA arrays and subjected the arrays to gentle washing using Agilent Technologies’ Gene Expression Wash Buffer as described in the manufacturer’s protocol. The arrays were then analyzed at 3 µm resolution using a microarray scanner (Agilent Technologies). The resulting data were analyzed using the Agilent Feature Extraction software program (ver. 12.0.3.1).

### 2.4. The Processing of the mRNA Microarray Data and the Statistical Analyses

We input the Agilent data into the Agilent Technologies software program GeneSpring GX (ver. 14.9.1); the data were then normalized to the 90th percentile per chip. Based on the data of the SAMR1-10wk mice, we performed a baseline correction. A one-way analysis of variance (ANOVA) was conducted to investigate between-group differences, and the post hoc analysis applied Tukey’s test. Probability (*p*)-values < 0.05 were accepted as significant [11]. Using the microarray analysis results, we obtained cluster and heat maps of the mRNAs that meet the conditions described herein.

### 2.5. Real-Time Quantitative Reverse-Transcription Polymerase Chain Reaction

The kidney specimens from the mice were homogenized using a silica homogenizer and a Qiagen filter column shredder (QIAshredder column). We used Qiagen’s RNeasy^®^ Mini Kit to extract mRNA from each kidney, and we then used the Super-Script^®^ III First-Strand Synthesis System (ThermoFisher Scientific, Waltham, MA, USA) to reverse-transcribe 1 µg of each extracted RNA. The qRT-PCR (quantitative real-time quantitative reverse-transcription polymerase chain reaction) was performed using SYBR^®^ GreenER™ qPCR SuperMix (ThermoFisher Scientific).

The following were purchased from Takara Bio (Otsu, Shiga, Japan): primers for mouse β-actin; acyl-Coenzyme A oxidase 2; branched chain (Acox2); apolipoprotein C-II (Apoc2); caspase recruitment domain family, member 11 (Card11); Cluster of differentiation (Cd) 19, Cd2, Cd37, Cd52, Cd79b, and Cd83; cartilage oligomeric matrix protein (Comp); dedicator of cyto-kinesis 2 (Dock2); deltex 1 (Dtx1); fibroblast-specific-protein-1 (FSP-1); interleukin (IL)-1b; lymphotoxin b (Ltb); matrix metallopeptidase (MMP3); neutrophil cytosolic factor 1 (Ncf1); neuropeptide Y receptor Y6 (Npy6r); nuclear receptor subfamily 4, group A, member 1 (Nr4a1); podocin; runt-related transcription factor 3 (Runx3); SAM domain; SH3 domain and nuclear localization signals, 1 (Samsn1); vimentin; and zeta-chain-associated protein kinase (Zap70).

The primers used were as follows: β-actin, Acox2, Card11, Cd19, Cd2, Cd37, Cd52, Cd79b, Cd83, Comp, Deck2, Dtx1, FSP-1, IL-1b, Ltb, MMP3, Npy6r, Nr4a1, Runx3, podocin, Samsn1, vimentin, and Zap70. The expression level of each mRNA was normalized in relation to that of β-actin. The results are presented in relation to the SAMR-10wk group.

### 2.6. Histological Analysis

At the indicated time points, the mice were sacrificed through cervical dislocation. Reflux flow with phosphate-buffered saline (PBS) was injected into left ventricle. Both kidneys were removed and fixed with 4% paraformaldehyde. The kidneys were embedded in paraffin and sectioned at 5 µm thickness; the sections underwent Azan staining for an assessment of fibrous changes. We applied PAS (periodic acid-Schiff) staining to determine the glomerulosclerosis index. We conducted a quantitative analysis of the fibrosis degree by examining ten fields per section that was positively stained with Azan, at 200× magnification. Randomly chosen fields were used for this analysis, and the Azan-stained areas were quantified using the image-analysis software (BZ-H3A ver. 1.4.1.1) program provided with the fluorescence microscope used (BZ-X710; Keyence, Osaka, Japan). To determine the sclerosis degree, we assessed 50 glomeruli that had been positively stained for PAS at 200× magn [2]. The glomerulosclerosis index values were defined as follows—1 point: no change in the glomeruli; 2 points: a proliferation of mesangial cells; 3 points: segmental sclerosis; 4 points: total glomerulosclerosis [2].

### 2.7. Statistical Analyses

The data obtained in all analyses are presented as the mean ± standard error (SE). Since the data were normally distributed in all groups, for the multiple-comparison analysis among groups, we performed an ANOVA on the basis of previous report [12]. When the ANOVA indicated statistical significance, we applied Tukey’s test as a post hoc analysis for a comparison of two groups’ means. The relationships among continuous variables were analyzed using Pearson’s correlation test or a linear regression analysis. We used JMP software ver. 13 (SAS Institute, Cary, NC, USA) for all statistical analyses. Probability (*p*)-values < 0.05 were accepted as significant.

## 3. Results

### 3.1. Kidney Weights in the SAMP1-50wk Mice

Figure 1A depicts the kidney weights in each group of mice. The kidney weight was notably downregulated in the SAMP1-50wk mice versus the SAMR1-50wk mice.

### 3.2. Renal Fibrosis and Glomerulosclerosis in the SAMP1-50wk Mice

The histological evaluation of the kidney samples revealed that unlike the SAMP1-10wk mice and the SAMR1-10wk and SAMR1-50wk mice, the kidneys from the SAMP1-50wk mice exhibited typical features of fibrosis and glomerulosclerosis (Figure 1B–E). In addition, the qRT-PCR results demonstrated increases in two renal fibrosis markers, i.e., vimentin and FSP-1, in the SAMP1-50wk group (Figure 1C,D) plus a decrease in the expression of podocin (a glomerulosclerosis marker) in these mice (Figure 1E).

### 3.3. Microarray mRNA Profiling

We investigated changes in mRNA expressions in age-dependent renal impairment in the mice by using a microarray platform that covers 62,976 mouse mRNAs, using GenBank (August 2014) (GenBank Overview, available at nih.gov); RIKEN 3 (RNA Systems Biochemistry Laboratory|RIKEN, RefSeq Build); RefSeq Build 66, the U.S. NCBI Reference Sequence Database (nih.gov); Unigene Build 236; and the Sol Genomics Network (SGN) Unigene Builds, Ensemble release 76, Ensembl genome browser 107.

The screening revealed that the expression of 44 of the 62,976 mRNAs was significantly different in the SAMP1-50wk mice compared to the SAMR1-10wk, SAMR1-50wk, and SAMP1-10wk groups through one-way ANOVA. The 44 mRNAs are as follows: 4921532D01Rik, 4931414P19Rik, A530032D15Rik, Acox2, Apoc2, basic leucine zipper ATF like transcription factor (Batf), Card11, Cd19, Cd2, Cd37, Cd52, Cd79b, Cd83, Ceacam16, centromere protein F (Cenpf), Comp, colony stimulating factor 2 receptor, beta 2 (Csf2rb2), component of Sp100-rs (Csprs), Dock2, Dtx1, Fc mu receptor (Fcmr), Gm10050, H2-Oa, IKAROS family zinc finger 3 (Ikzf3), IL-1b, LOC664787, Ltb, lysozyme 1 (Lyz1), mir142 host gene (Mir142hg), MMP3, membrane-spanning four-domains, subfamily A, member 1 (Ms4a1), neutrophil cytosolic factor 1 (Ncf1), Npy6r, Nr4a1, Protocadherin 7 (Pcdh7), phospholipase A2 group IID (Pla2g2d), Runx3, S100 calcium binding protein G (S100g), Samsn1, serine (or cysteine) peptidase inhibitor, clade A, member 3N (Serpina3n), solute carrier family 7 member 12 (Slc7a12), tumor necrosis factor receptor superfamily member 13c (Tnfrsf13c), WAP four-disulfide core domain 3 (Wfdc3), and Zap70 (Appendix A, Figure 2).

We also detected 44 mRNAs, the expressions of which were increased by more than twofold or decreased by >50% in the SAMP1-50wk mice compared to the other three groups of mice: 4921532D01Rik, 4931414P19Rik, A530032D15Rik, Acox2, Apoc2, Batf, Card11, Cd19, Cd2, Cd37, Cd52, Cd79b, Cd83, Ceacam16, Cenpf, Comp, Csf2rb2, Csprs, Dock2, Dtx1, Fcmr, Gm10050, H2-Oa, Ikzf3, IL1b, LOC664787, Ltb, Lyz1, Mir142hg, MMP3, Ms4a1, Ncf1, Npy6r, Nr4a1, Pcdh7, Pla2g2d, Runx3, S100g, Samsn1, Serpina3n, Slc7a12, Tnfrsfl3c, Wfdc3, and Zap70.

In the SAMP1-50wk group, the mRNAs 4921532D01Rik, 4931414P19Rik, Acox2, Cenpf, Gm10050, Npy6r, Pcdh7, S100g, and Wfdc3 were expressed at levels lower than 50% of the levels in the SAMR1-10wk, SAMP1-10wk, and SAMR1-50wk mice (Appendix A, Figure 2). Before our statistical analysis of the microarray data, a flag filter (i.e., 100% present in at least one group) was used by the GeneSpring software ver. 14.9.1, and 23 mRNAs were then excluded from the analysis: 4921532D01Rik, 4931414P19Rik, A530032D15Rik, Apoc2, Batf, Ceacam16, Cenpf, Csf2rb2, Csprs, Fcmr, Gm10050, H2-Oa, Ikzf3, LOC664787, Lyz1, Mir142hg, Ms4a1, Ncf1, Pcdh7, Pla2g2d, S100g, Serpina3n, Slc7a12, Tnfrsfl3c, and Wfdc3.

We confirmed the expression levels of 21 miRNAs in the four mouse groups using qRT-PCR: Acox2, Card11, Cd19, Cd2, Cd37, Cd52, Cd79b, Cd83, Comp, Deck2, Dtx1, IL1b, Ltb, MMP3, Npy6r, Nr4a1, Runx3, Samsn1, and Zap70. As illustrated in Figure 3, the results of this analysis demonstrated significantly upregulated expression levels of five mRNAs (Cd37, Comp, Dtx1, IL-1b, and Nr4a1) in the SAMP1-50wk mice versus the control, i.e., SAMR1-10wk, SAMR1-50wk, and SAMP1-10wk mice. No mRNAs in the SAMP1-50wk mice were downregulated compared to the control groups (Figure 3).

## 4. Discussion

This study revealed that the kidney expressions of the five mRNAs, Cd37, Comp, Dtx1, IL-1b, and Nr4a1, were increased in mice that have age-dependent renal impairment, indicating that these mRNAs might be related to age-dependent renal impairment (Appendix A, Figure 4). The mRNA Cd37 is a molecule that is expressed on the surface of B cells, and it is reported to be involved in the development of immunoglobulin A nephropathy with renal fibrosis and glomerulosclerosis in Cd37-deficient mice [13]. In our present investigation, the expression level of Cd37 was elevated in the mice with age-dependent renal impairment, suggesting that Cd37 expression is increased as protection against the development of age-dependent renal impairment. It is necessary to further investigate the mechanism by which Cd37 may suppress renal fibrosis and glomerulosclerosis.

Nr4a1 is one of the nuclear receptors and is reported to be upregulated in mice podocytes in hyperglycemia [14]. Nr4a1 was observed to promote renal fibrosis in mice by activating tumor growth factor β-signaling and increasing collagen I/III/IV and MMP3 [14]. In addition, Nr4A1 is reported to promote diabetic nephropathy by enhancing glomerulosclerosis via activation of the p53 signaling pathway and mitochondrial apoptosis in podocytes [14]. The same study revealed that Nr4a1 deficiency inhibits renal fibrosis and improves diabetic nephropathy in streptozotocin-induced diabetic nephropathy model mice [14]. In our present investigation, the expression level of Nr4a1 was elevated in the mice with age-dependent renal impairment. This suggests that Nr4a1 promotes renal fibrosis and glomerulosclerosis and may be a potential factor in the development of not only diabetic nephropathy but also age-dependent renal impairment.

The pentameric glycoprotein Comp (also known as thrombospondin 5) is expressed in tissues such as fibroblasts/myofibroblasts, myocardial cells, vascular smooth muscle cells, cells of various tumor types, cartilage, the synovium, and ligaments [15]. Comp is also active in collagen secretion and fibril formation and has been shown to interact with other extracellular matrix components such as collagen (types I, II, and IX), matrilin 3, aggrecan, and fibronectin [16,17]. It was reported that Comp was elevated in the serum of older adults with renal failure [18]. The expression level of Comp was elevated in the present study’s mouse model of age-dependent renal impairment. This result is consistent with reports that Comp could be used as a biomarker in the development of age-dependent renal impairment. The mechanism by which Comp increases in kidneys with age-dependent renal impairment remains to be clarified in further studies.

IL-1b is a pro-inflammatory cytokine and has been described as a marker of renal fibrosis [19,20]. We observed herein that the expression of IL-1b was upregulated in mouse kidneys with age-dependent renal impairment. This result suggested that IL-1b is elevated in age-dependent renal impairment through renal fibrosis caused by an inflammation response.

A deficiency of Dtx1 was reported to promote T-cell activation and increased inflammation [21]. Dtx1 is an important signaling transcriptional regulator downstream of the Notch receptor [22], and it was reported that Notch inhibition leads to the amelioration of glomerulonephritis in a murine model of systemic lupus erythematosus by inhibiting the inflammation response [23]. Our present results demonstrated that Dtx1 was upregulated in age-dependent renal impairment. We speculate that Dtx1 may be upregulated to suppress the inflammation response in the kidney in age-dependent renal impairment. However, further investigation is needed to establish the mechanism by which age-dependent renal impairment is caused through an inflammation response.

Taken together, our present findings demonstrate that the kidney expression levels of the five mRNAs, Cd37, Comp, Dtx1, IL-1b, and Nr4a1, were increased in mice with age-dependent renal impairment. These results indicate that the five mRNAs might be related to age-dependent renal impairment, and that they might be therapeutic targets for age-dependent renal impairment.

Some study limitations should be addressed. The analyses of our results might have been affected by the sample sizes. The expressions of the mRNAs were not validated in serum or plasma. The amounts of protein of the molecules were not evaluated using Western blotting. It has not been verified whether these mRNAs can become therapeutic targets for age-dependent renal impairment. We plan to address these limitations in future studies.

## 5. Conclusions

Our present findings demonstrate that kidney levels of Cd37, Comp, Dtx1, IL-1b, and Nr4a1 were increased in mice with age-dependent renal impairment, indicating that these mRNAs might be related to age-dependent renal impairment. Furthermore, therapeutic effects can be expected through overexpression of these mRNAs or the creation of knockout mice.

## Figures and Tables

**Figure 1 diagnostics-13-03653-f001:**
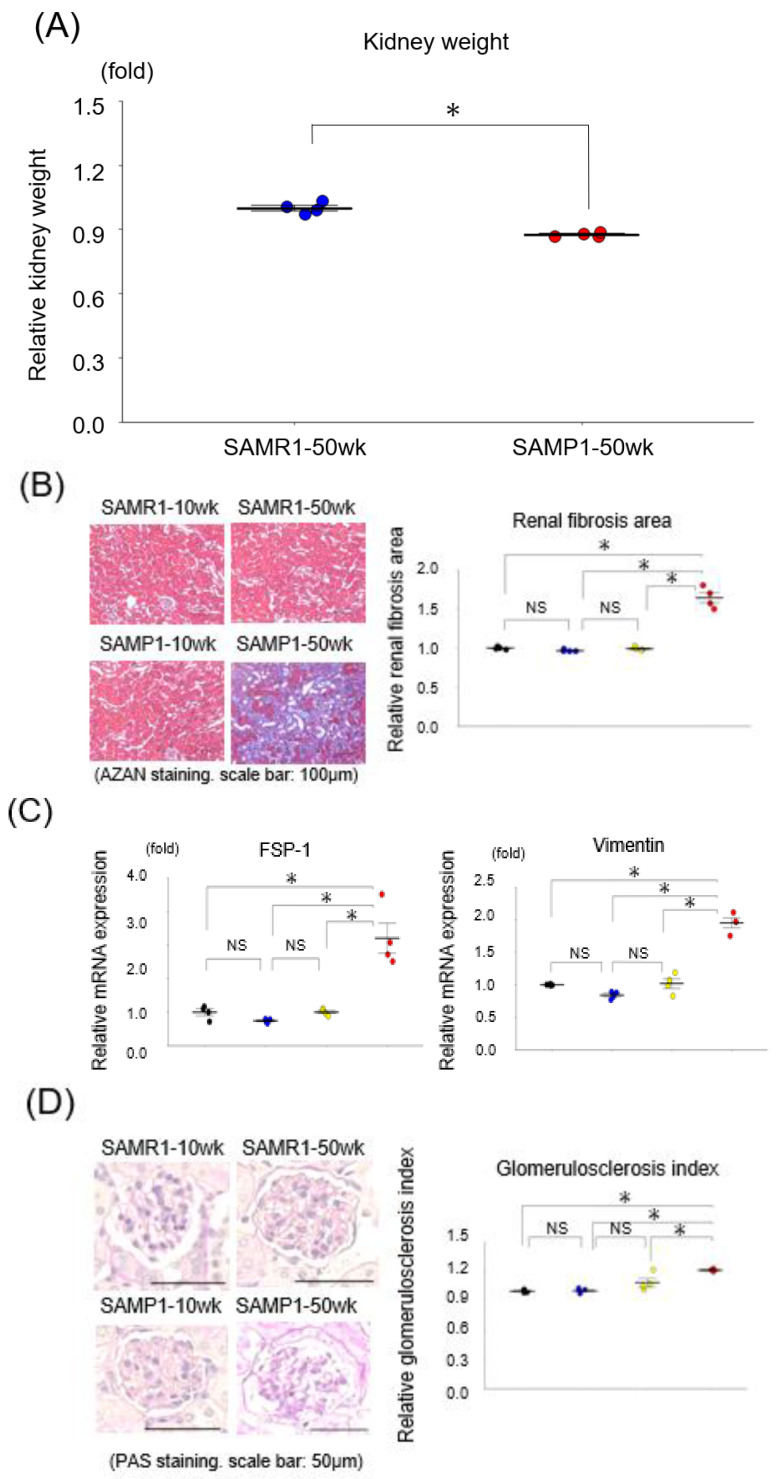
Changes in interstitial nephropathy and glomerulosclerosis in the SAMR1-10wk group, SAMR1-50wk group, SAMP1-10wk group, and SAMP1-50wk group. (**A**) Kidney weights in the SAMR1-50wk group and SAMP1-50wk group. (**B**) Azan staining paraffin sections. Magn.: ×200. Scale bar = 100 μm. We observed marked renal fibrosis in the SAMP1-50wk group. The changes in renal fibrosis area in each group (*n* = 4) are shown. (**C**) The results of the qRT-PCR analysis of FSP-1 and vimentin expression in the groups (*n* = 4 each). (**D**) PAS-stained paraffin sections; magn. ×200. Scale bar + 50 μm. We observed marked glomerulosclerosis in the SAMP1-50wk group. The changes in the glomerulosclerosis index in the groups (*n* = 4 each) are shown. (**E**) The results of the qRT-PCR analysis of podocin expression in the groups (*n* = 4 each) are shown. Values are mean ± SE (indicated by error bars). NS, not significant. * *p* < 0.05, ANOVA, Tukey’s test. FSP-1: fibroblast-specific-protein 1.

**Figure 2 diagnostics-13-03653-f002:**
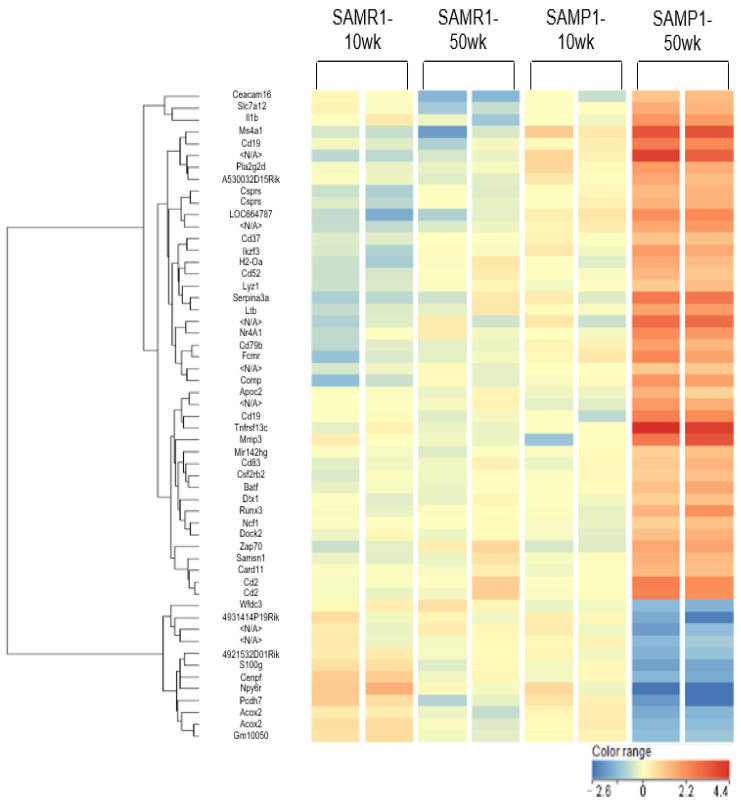
The heat map of the mRNAs that were differentially expressed in the kidneys of the SAMR1-10wk mice, SAMP1-10wk mice, SAMR1-50wk mice, and SAMP1-50wk mice. The hierarchical clustering and systemic variations in the mRNA expressions in the mouse kidneys in each group (*n* = 4 each) are shown. The expression of the mRNAs presented in red was high, and that of the mRNAs presented in green was relatively low. The figure’s color version is available online.

**Figure 3 diagnostics-13-03653-f003:**
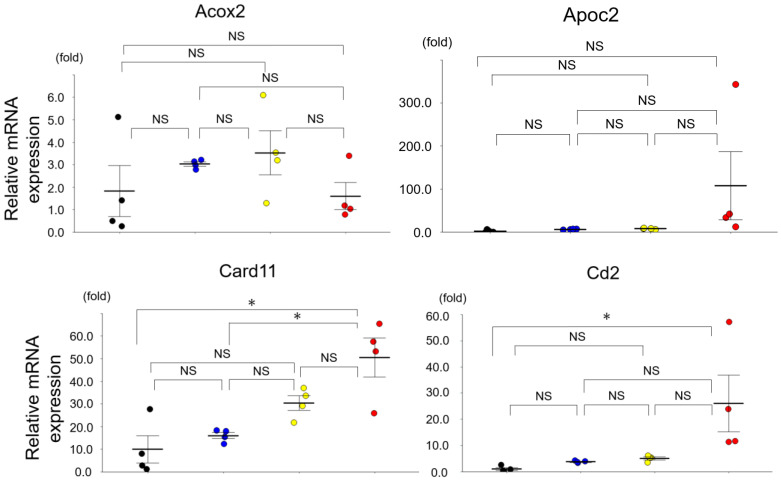
The expressions of mRNAs revealed by a qRT-PCR in the mouse kidneys. The mRNA expressions of Cd37, Comp, Dtx1, IL-1b, and Nr4a1 in the SAMP1-50wk mice were upregulated compared to those in the SAMR1-10wk, SAMR1-50wk, and SAMP1-10wk mice (each group, *n* = 4). Values are mean ± SE (error bars). * *p* < 0.05, ANOVA, Tukey’s test. Acox2: acyl-Coenzyme A oxidase 2, branched chain; Apoc2: apolipoprotein C-II; Card11: caspase recruitment domain family, member 11; Comp: cartilage oligomeric matrix protein; Dock2: dedicator of cyto-kinesis 2; Dtx1: deltex 1; IL-1b: interleukin-1b; Ltb: lymphotoxin b; MMP3: matrix metallopeptidase3; Ncf1: neutrophil cytosolic factor 1; Npy6r: neuropeptide Y receptor Y6; Nr4a1: nuclear receptor subfamily 4, group A, member 1; Runx3: runt-related transcription factor 3; Samsn1: SAM domain, SH3 domain and nuclear localization signals, 1; Zap70: zeta-chain-associated protein kinase. NS, not significant.

**Figure 4 diagnostics-13-03653-f004:**
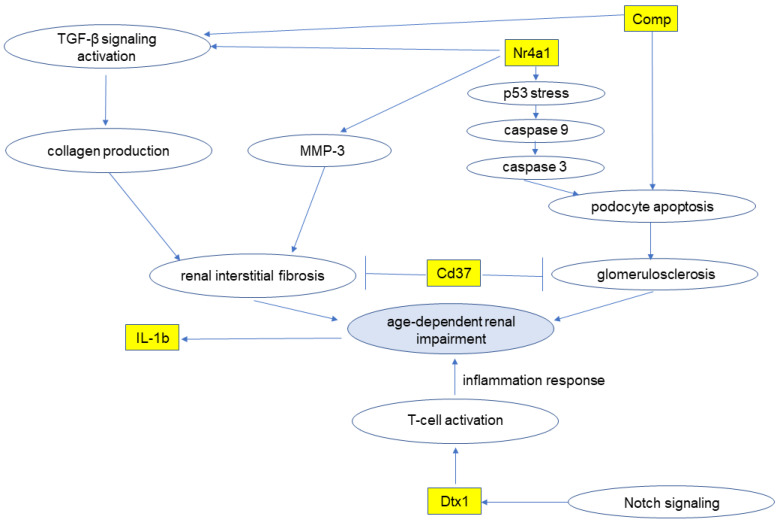
Age-dependent renal impairment develops via various pathways through five mRNAs. Cd: cluster of difference; Comp: cartilage oligomeric matrix protein; Dtx1: deltex 1; IL-1b: interleukin-1b; MMP3: matrix metalloprotease3; Nr4a1: nuclear receptor subfamily 4, group A; TGF: tumor growth factor.

## Data Availability

Publicly available datasets were analyzed in this study. This data can be found here: [https://www.ncbi.nlm.nih.gov/geo/query/acc.cgi?acc=GSE249488] (accessed on 7 November 2023).

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
