# Peer review of "Identification of Messenger RNA Signatures in Age-Dependent Renal Impairment"

_diagnostics, 2023, doi:10.3390/diagnostics13243653_

Round 1

Reviewer 1 Report

Comments and Suggestions for Authors

Dear authors, 

I have some comments about your paper:

-improve introduction

-explicit aim of the study  in the paper

-specify how many animal did you use? I suppose one animal for type: so your data have to be validated and confirmed

-add limits of the study

-why did you use ANOVA and parametric tests? did you test data?

the work is very interesting but the data have be to validate. Conclusions are hypothesis-genereting data and are to confirmed

Comments on the Quality of English Language

English is ok

Reviewer 2 Report

Comments and Suggestions for Authors

The paper is meticulously structured and logically organized. It effectively integrates pertinent and recent studies within the References section. Ethical guidelines were diligently followed, ensuring the proper treatment of the animals involved. The discussion section aptly identifies and deliberates upon the study's limitations. Despite certain ambiguities and the necessity for further investigation, this paper serves as a promising foundation for future research endeavors.

Round 2

Reviewer 1 Report

Comments and Suggestions for Authors

This version is ok for publication